# How Dexamethasone Used in Anti-COVID-19 Therapy Influenced Antihypertensive Treatment in Patients with SARS-CoV-2

**DOI:** 10.3390/healthcare11101399

**Published:** 2023-05-11

**Authors:** Andrei Puiu Cârstea, Adrian Mită, Mircea-Cătălin Fortofoiu, Irina Paula Doica, Doina Cârstea, Cristina Maria Beznă, Cristina Elena Negroiu, Ileana-Diana Diaconu, Andreea-Roberta Georgescu, Adina Maria Kamal, Beatrice Mahler, Adriana-Gabriela Grigorie, Gabriel Adrian Dobrinescu

**Affiliations:** 1Department of Physiology, Faculty of Medicine, University of Medicine and Pharmacy of Craiova, 200349 Craiova, Romania; 2Department of Cardiology, Clinical Municipal Hospital “Philanthropy” of Craiova, 200143 Craiova, Romania; 3Department of Medical Semiology, Faculty of Medicine, University of Medicine and Pharmacy of Craiova, 200349 Craiova, Romania; 4Department of Internal Medicine 2—Gastroenterology Compartment, “Philanthropy” Clinical Municipal Hospital of Craiova, 200143 Craiova, Romania; 5Doctoral School, University of Medicine and Pharmacy of Craiova, 200349 Craiova, Romania; 6Department of Cardiology, Clinical County Emergency Hospital of Craiova, 200642 Craiova, Romania; 7Department of Pediatric Pneumology, National Institute of Pneumology “Marius Nasta” of Bucharest, 050159 Bucharest, Romania; 8Department of Emergency Medicine, Clinical County Emergency Hospital of Craiova, 200642 Craiova, Romania; 9Department of Internal Medicine, Faculty of Midwives and General Nursing, University of Medicine and Pharmacy of Craiova, 200349 Craiova, Romania; 10Department of Internal Medicine 1, “Philanthropy” Clinical Municipal Hospital of Craiova, 200143 Craiova, Romania; 11Department of Pneumology, University of Medicine and Pharmacy “Carol Davila” Bucharest, 020021 Bucharest, Romania; 12National Institute of Pneumophtisiology “Marius Nasta”, 050159 Bucharest, Romania; 13Leamna Pneumophtisiology Hospital, Leamna de Sus, 207129 Dolj, Romania; 14Department of Thoracic Surgery, Faculty of Medicine, University of Medicine and Pharmacy of Craiova, 200349 Craiova, Romania; 15Department of Thoracic Surgery, Clinical County Emergency Hospital of Craiova, 200642 Craiova, Romania

**Keywords:** dexamethasone, SARS-CoV-2 infection, blood pressure, COVID-19 pandemic

## Abstract

Background: During the SARS-CoV-2 pandemic period, in the treatment approved by the WHO, along with antivirals, antibiotics, nonsteroidal anti-inflammatory drugs and anticoagulants, dexamethasone was always used. This study started from the professional concern related to the vasopressor effect of cortisone on blood pressure (BP). Methods: The study group was achieved by selecting, from a total of 356 patients hospitalized in the clinic, the patients with known hypertensive status at admission for SARS-CoV-2. Dexamethasone was part of the anti-COVID-19 treatment, with an administration of 4–6–8 mg/day, depending on bodyweight, for 10 days. All patients with hypertension received antihypertensive treatment in adjusted doses according to the recorded BP values. Results: Monitoring of BP in hospitalized patients was performed daily, in the morning and evening. If on the 2nd day of treatment, 84% of the patients partially responded to the treatment with a moderate decrease in BP, on the 3rd therapy day, the situation clearly improved: more than 75% of the patients had values of BP that can be classified as high-normal (38.23%) and normal (40.03%). Conclusions: Dexamethasone for treatment of SARS-CoV-2 infection did not have a notable influence on increasing BP, because the doses were low–moderate and prescribed for a short time.

## 1. Introduction

During the SARS-CoV-2 pandemic (2020–2021), in all the treatment schemes approved by the WHO and taken over worldwide by all countries, including the USA and the EU, along with antivirals, antibiotics, nonsteroidal anti-inflammatory drugs and anticoagulants [1,2,3], dexamethasone was always used. 

In the first months after the pandemic outbreak, especially in the first half of 2020, therapeutic schemes launched by infectionists [4,5,6,7,8,9], defined as being the most suitable for the moment, varied a lot. Of all the immunomodulatory drugs, only tocilizumab was available for 2–3 months towards the end of 2020, and only in intensive care units; otherwise, dexamethasone was widely used. Paracetamol was also used in anti-COVID-19 therapeutic management throughout the pandemic and was classified in the “other therapeutic measures” class, along with antitussives, broncholytics and bronchodilators [10].

In the present study, only drugs that were available and used in the EU and in Romania, respectively, are listed [11]. In the studied pandemic period, hospital admission for diseases other than SARS-CoV-2 was drastically restricted (with the exception of major medical–surgical emergencies). Learning and adapting to everything from personal equipment to extremely strict hygiene rules, and especially to the treatment and monitoring of a condition that was new and different from our specialty, was frustrating [12]. 

The condition, often serious, of cardiac and COVID-19 patients meant that all our attention was directed towards the most correct application of the SARS-CoV-2 treatment protocol, along with the follow-up of each patient admitted to the cardiology clinic [13]. Only after we gained some experience and adapted to the new drugs did we observe improve and ask questions so that we could achieve some early conclusions [14,15].

In the present study, we assess the evolution of arterial hypertension (HBP—high blood pressure) in COVID-19-positive patients admitted to the clinic, due to the fact that dexamethasone was always included in the anti-COVID-19 treatment management. The study started, as mentioned, from the professional “fear” of not encountering medical problems related to the influence (well known in the medical world) of cortisone preparations on blood pressure level [16]. It is known that the majority of steroid pharmaceutical derivatives (injectable glucocorticoid preparations especially) have an intense and sustained vasopressor effect, leading to significant increases in blood pressure values [16,17]. Dexamethasone is a synthetic glucocorticoid that is used as an anti-inflammatory and immunosuppressive agent. While it has numerous beneficial effects, one of its potential side effects is hypertension or high blood pressure. The exact mechanism by which dexamethasone increases blood pressure is not fully understood, but several theories have been proposed [18,19]:Sodium retention: One of the most commonly accepted theories is that dexamethasone increases sodium retention by the kidneys. This leads to an increase in extracellular fluid volume and subsequently an increase in blood pressure.Vasoconstriction: Dexamethasone has been shown to cause vasoconstriction, or narrowing of the blood vessels, in some animal studies. This can lead to an increase in systemic vascular resistance and subsequently an increase in blood pressure.Sympathetic nervous system activation: While dexamethasone does not directly activate the sympathetic nervous system, it can indirectly affect it by reducing inflammation and suppressing the immune system. This response involves the release of various hormones, including adrenaline, which increase heart rate, blood pressure and breathing rate, among other physiological changes.

## 2. Materials and Methods

Our observational study was conducted between 1 July 2020 and 30 June 2021 on a group of 356 patients admitted to the Cardiology Clinic of Philanthropy Clinical Municipal Hospital in Craiova—A COVID-19 support hospital unit (Order no. 555/2020). Thus, hospitalization in all of this hospital’s clinics (and implicitly in the cardiology one) was only allowed for patients infected with SARS-CoV-2 who were also associated with other conditions and whose severity at the time of hospitalization was high and could not receive appropriate care only in infectious diseases units in the neighboring areas [20]. This study was prepared in accordance with the provisions of the Helsinki Declaration from 1964 (The 18th World Medical Assembly), revised within the framework of the 29th World Medical Assembly from Tokyo in 1975. We respected the WHO principles regarding patients’ rights and the Law on Patients’ Rights 46/2003. Moreover, the study received favorable approval from the Ethics Committee of the Municipal Philanthropy Clinical Hospital (no. 18798/19.10.2021).

The study group was achieved by selecting, from the total number (356) of patients admitted to the clinic during the observed period, only the patients known to be hypertensive or with acute hypertensive episodes at admission due to SARS-CoV-2 infection. The rest of the patients with other previously known heart conditions and who presented blood pressure (BP) values below 160 mmHg systolic at presentation were not included in the group. It is well known that there is no well-defined “line” between cardiac diseases, but the basic criteria for including or excluding patients in the group was provided by HBP as reason for hospitalization. 

Anti-COVID-19 treatment included antibiotics, anticoagulants, antivirals and dexamethasone [21] (as immune-modulator), as well as paracetamol and other symptomatic drugs, in doses and combinations prescribed by the infectious disease doctors accredited to our hospital [22]. Dexamethasone was administered for 10 days in doses up to the maximum recommended by the EMA (European Medicines Agency) of 6 mg per day for normal-weight adults, but it required dosage adaptation for patients with degree III obesity and above (according to the CHMP (Committee for Medicinal Products for Human uses)). Dexamethasone was indicated and was part of the treatment regimens recommended by the WHO for the treatment of SARS-CoV-2 infections [23]. The WHO updated its COVID-19 treatment guidelines in September 2020 to include the use of dexamethasone for the treatment of patients with severe and critical COVID-19. Since then, dexamethasone has become a widely used medication in the treatment of severe COVID-19 cases and is typically administered as part of a multimodal treatment plan that may include other medications, oxygen therapy and supportive care. Dexamethasone is a corticosteroid that has anti-inflammatory properties and can suppress the immune system. It was found to be effective in reducing mortality in patients with severe COVID-19 infection in the RECOVERY trial, a large-scale clinical trial conducted in the United Kingdom [17]. The study found that dexamethasone reduced the risk of death by one-third in patients receiving mechanical ventilation and by one-fifth in patients receiving oxygen therapy but not on a ventilator. The drug did not have a significant effect on patients with mild or moderate COVID-19 infection.

Based on these results, the World Health Organization (WHO) recommended the use of dexamethasone in patients with severe and critical COVID-19 infection who require oxygen therapy or mechanical ventilation. The drug is not recommended for use in patients with mild or moderate COVID-19 infection [24]. It is important to note that dexamethasone should only be used under the supervision of a healthcare professional, as it can have side effects such as increased blood sugar, weight gain, and increased risk of infection. The use of dexamethasone in the treatment of COVID-19 may have implications for patients with underlying cardiovascular conditions, including hypertension, and the elderly population. Dexamethasone is known to have potential side effects on blood pressure, glucose metabolism and fluid balance. Specifically, it can cause a transient increase in blood pressure due to sodium and water retention, which can be more pronounced in patients with pre-existing hypertension. Therefore, caution should be exercised when administering dexamethasone to patients with underlying cardiovascular conditions, including hypertension, as it may exacerbate their condition. Close monitoring of blood pressure and electrolyte levels is recommended in these patients. In addition, the elderly population is more susceptible to the adverse effects of dexamethasone due to age-related changes in their metabolism and organ function [20]. Therefore, the use of dexamethasone in elderly patients should be carefully evaluated and monitored by a healthcare professional.

Overall, while dexamethasone has been shown to be an effective treatment for severe COVID-19, its use should be approached with caution in patients with underlying cardiovascular conditions and the elderly population, and it should only be used under the supervision of a healthcare professional.

Blood pressure was measured in the morning and evening, as well as for specific symptoms such as headache, dizziness, vertigo and palpitations. In the context of the infectious disease, these symptoms were either difficult to attribute entirely to hypertension or (even if they were caused by high blood pressure values) were partially masked by the dominant symptoms of the COVID-19 infection: dyspnea, fever, cough, asthenia, palpitations, precordialgias and anosmia.

The phrase should be reformulated as follows:Optimal—SBP < 120 mmHg and DBP < 80 mmHg;Normal—SBP 120–129 mmHg and/or DBP 80–84 mmHg;High-normal—SBP 130–139 mmHg and/or DBP 85–89 mmHg;Grade 1—SBP 140–159 mmHg and/or DBP 90–99 mmHg;Grade 2—SBP 160–179 mmHg and/or DBP 100–109 mmHg;Grade 3—SBP ≥ 180 mmHg and/or DBP ≥ 110 mmHg;Isolated systemic hypertension—SBP ≥ 140 mmHg and DBP < 90 mmHg. 

Legend: SBP, systolic blood pressure; DBP, diastolic blood pressure.

In patients with high BP, we recorded blood pressure values through morning and evening measurements and also when specific symptoms occurred. As antihypertensive treatment, we only used diuretics, calcium channel blockers (CCB), angiotensin-converting enzyme inhibitors (ACE), angiotensin receptor blockers (ARB) and central agonist of alpha 2 imidazolinic receptors (rilmenidine) in doses adjusted to the recorded BP values (according to the last JNC) [25,26,27]. The charts that illustrate the evolutional trend of the various evaluated parameters, as well as the statistical analysis between them, were achieved using the “Graph” tool of “Word” and “Excel” modules of the Microsoft Office 2016 Professional software package and the program type “Add on” XLSTAT for “Excel” module trial version. 

## 3. Results

For the 356 patients with SARS-CoV-2 infections who were admitted between 1 July 2020 and 31 June 2021 to the cardiology clinic, the main cardiologic diagnosis (as the reason for admission) is shown in Figure 1 and Figure 2, where it is easy to see that HBP was the reason for hospitalization in 238 patients (67%, with a difference of 5% more for the female sex). We also noted that the age interval of 66–75 years represented a percentage of 43.27%, with females having 11 cases more than males. The 2nd-ranked age group was 56–65 (27.73%), where more males were found, followed by the 76–85 age group, with a percentage of 17.64%, with a slightly higher number of females.

These hypertensive patients at the time of hospitalization, regardless of their own medication at home or those unmedicated until that moment, received diuretic treatment (1 cp/day: furosemide 20 mg, spironolactone 50 mg), CCB (1 cp/day amlodipine 10 mg–5 mg), ACE (1 cp/day perindopril 10 mg–5 mg) and ARB (candesartan 16 mg/day), and in some cases, rilmenidine 1 mg/day. These doses were adjusted according to BP value. 

We also mention that the antihypertensive medication scheme was initially simple, with the intention of seeing the patient’s response to treatment and to be able to subsequently increase the medication by adding other classes of antihypertensives, or to decrease it depending on the evolution. Among other cardiological conditions, atrial rhythm disorders reached a percentage of 10.67%, followed by ischemic heart disease at 8.70%. 

Blood pressure (BP) values and evolution were monitored daily in the enrolled patients, in the morning and evening. The graphic representation of their response to treatment was achieved on the 2nd, 3rd and 7th day upon admission, as well as on on the 14th day, before discharge. We mention that antihypertensives doses were adjusted according to individual BP values, and we also note that for 10 days, the patients were medicated with dexamethasone at an average dose of 4–6–8 mg depending on their body weight and the severity of the disease (as recommended by the infectionist). 

If we study the BP values at admission (Figure 3), systolic values higher than 200 mmHg were only recorded in 6 patients (2.5%), which is explained by the fact that the patients who arrived at the emergency unit were already medicated by the ambulance medical team. Most patients (105, representing 44.1%) recorded values between 189–180/95–90 mmHg when measuring BP in the emergency room, followed by 32.3% with values between 179–160/90–85 mmHg. Systolic BP values of about 190 mmHg were recorded in 21.1% of cases. 

Hospitalized patients immediately received the medication from the anti-COVID-19 protocols, and therefore also dexamethasone, in doses adjusted according to their bodyweight but not exceeding the value of 8 mg/day, for 10 days. In addition to the treatment recommended by the infectionist, we also instituted antihypertensive treatment as mentioned previously with diuretics, CCB, ACE and ARB. 

After initiation of classical antihypertensive treatment, at the first blood pressure measurement, in the morning or evening, depending on the time of day when the hospitalization occurred, it can be noticed in Figure 4 that arterial blood pressure values were significantly reduced in 200 patients (84%) to values between 150–140/85–80 mmHg. In 13.5% of the patients, the values decreased to 179–160 mmHg. In a small number of only 6 patients (2.5%), blood pressure values remained higher at about 189–180/95–90 mmHg. 

For the latter, we decided to increase the diuretic dose and to double the ACE dose.

We decided not to intervene with strong antihypertensive therapy because some patients also had other cardiological medication with hypotensive effects, such as beta blockers, and in addition, a significant part of them were still febrile, with intravenous therapy either to correct dehydration or as support for antibiotic, antipyretic and antiviral treatment. Since the results obtained after only one day of treatment are not relevant for the evolution of the studied patients, we present in Figure 5 the BP values at 3 days after hospitalization and treatment. 

On the 2nd day of treatment, 84% of the patients partially responded to the treatment with a moderate decrease in BP, and 16% had more important decreases in BP (Figure 3)—although the results were not exactly what was expected. On the 3rd day of treatment, however, the situation was clearly improved: more than 75% of the patients presented BP values that can be classified as high-normal (38.23%) and normal (40.03%). Only a small part of the group’s patients, approximately 2%, maintained values above 160/90 mmHg, of which only 1 (0.42%) with 175/92 mmHg (a young person—55 years old with obesity, diabetes and severe dyslipidemia). 

The next data representation is from the 7th day of admission (Figure 6), a period in which we adjusted the antihypertensives doses according to the patients’ response. It was not necessary to add other categories of hypotensive drugs in any of the cases. 

From the chart that represents BP values on the 7th day of anti-COVID-19 (with dexamethasone) and antihypertensive treatment, it is noticed that values were closer to normal in almost half of the patients of our group (49.16%), and only in a small part (9.4%), the BP values remained at a slightly higher value (150–140/85–80 mmHg.) We also mention that 10 patients (4.2%) recorded decreases in BP values up to the inferior limit of normal (110–100/70–60 mmHg), although they were under antihypertensive medication at minimal doses of diuretic (indapamide 1.5 mg), CCB (amlodipine 2.5 mg) and ACE (perindropil 2.5 mg); thus, the diuretic from their medication was removed. All patients in the group received dexamethasone for 7 days in doses of 4–6–8 mg/day. 

Among the hypertensive patients, after 2 weeks of hospitalization (Figure 7), 67.22% had normal values under minimal diuretic–vasodilator antihypertensive treatment, the doses being adapted to each individual patient depending on their evolution. Another 77 patients (32.35%), had BP values normale or slightly elevated (129–120/80–75 mmHg), and almost all of them (91.78%) had other associate clinical conditions like these: severe heart diseases (50.1%), diabetes (32.56%: 87.3% with oral antidiabetic treatment and 12.7% with insulin intake) and obesity (15.16%). A single elderly patient (0.42%), although without antihypertensive medication from the 3rd day onwards (with dexamethasone 8 mg/day for 3rd degree obesity and severe COVID-19 manifestations), maintained low BP values despite intravenous hydration and the administration of cortisone support until discharge. However, it is important to note that maintaining low blood pressure values may not always be the desired outcome in patients with hypertension.

We mention that we did not record any paroxysmal increases in BP values in our hypertensive patients throughout their hospitalization during one year of study, although they had cortisone treatment with dexamethasone 4–6–8 mg/day for 10 days, and consequently, we did not experience any hypertensive emergency complications, such as hypertensive encephalopathy, cerebral edema, acute pulmonary edema and ischemic or hemorrhagic strokes. Hypertensive emergency is a serious medical condition that occurs when blood pressure levels rise to dangerously high levels. This can lead to a range of complications, including those mentioned above. 

## 4. Discussion

The general observation of the study shows that the evolution of hypertensive patients with COVID-19 respiratory infection, at the time of emergency presentation, did not consist of any particular problems regarding the response to classic antihypertensive therapy. The patients had a normal evolution, with a slow decrease in high blood pressure values a few days after admission, and towards the end of the first week of hospitalization, their BP values approached and remained normal. Thus, we cannot say that we faced increased BP values that may be attributed to ongoing dexamethasone treatment. On the contrary, in the first week of hospitalization, in about the 3rd and 4th day of treatment, the drop in BP values was more important than we would have expected, and we had to reduce the doses of antihypertensive medication, and in some patients, the treatment was interrupted. This may suggest that hypertension is not a major barrier to treating COVID-19 respiratory infection with standard antihypertensive therapy. However, it is important to note that hypertension can be a risk factor for severe COVID-19 disease and poor outcomes, as it is associated with increased risk of cardiovascular complications and other comorbidities [28]. In some cases, the management of hypertension in patients with COVID-19 may need to be adjusted based on individual patient circumstances and response to treatment. Additionally, the treatment of hypertension in patients with COVID-19 may need to take into account other factors related to the infection, such as the potential for drug interactions with other medications used in COVID-19 treatment, or the impact of respiratory symptoms on blood pressure control [29]. Close monitoring of blood pressure and other vital signs, as well as communication between healthcare providers involved in the patient’s care, can help to ensure that hypertension is effectively managed in the context of COVID-19 treatment [30]. The fact that the use of dexamethasone was or was not able to prevent these complications suggests that it may have had a positive impact on blood pressure regulation in the patients being treated [31]. We tried not to lower the BP values too much or too suddenly from the beginning, and we tried to keep them within normal parameters so as not to cause cerebral, coronary, renal or systemic hypoperfusion in this category of febrile and dehydrated patients [32]. The results of our study represent an explanation related to the fact that hypertension was the reason for hospitalization in two thirds (66.9%) of the studied patients, due to the emotional stress and anxiety related to the diagnosis of SARS-CoV-2. This severe diagnosis during the pandemic period and the confirmation of a disease with a possibly fatal evolution determined the occurrence of increased blood pressure values in patients previously without any medical history or led to the onset of acute hypertensive episodes in those with chronic treatment for several years [32]. It has been observed that SARS-CoV-2 infection can lead to an increase in blood pressure levels in some patients. This effect is thought to be due to the virus’s impact on the renin–angiotensin–aldosterone system (RAAS), which plays a crucial role in regulating blood pressure. The virus enters cells using the ACE2 receptor, which is also involved in regulating the RAAS. By binding to ACE2, the virus disrupts the balance of the RAAS, leading to an increase in the production of angiotensin II, a potent vasoconstrictor that can raise blood pressure levels. Moreover, COVID-19 can also lead to other complications that can contribute to high blood pressure, such as inflammation, endothelial dysfunction and thrombosis. It is essential to closely monitor blood pressure levels in COVID-19 patients, especially those with a history of hypertension. Proper management of blood pressure is crucial to reducing the risk of complications and improving outcomes for patients. An argument in favor of this reason is the fact that once admitted in the hospital, the medication, as well as the relief of fever, cough, asthenia and especially dyspnea, created a climate of comfort and trust for the patients, who began to respond favorably to antihypertensive medication. Moreover, fever and sweating, hyperventilation and implicitly dehydration (corrected by intravenous medication) maintained normal BP values and kept them constant throughout the hospitalization. In addition, more than half of the hospitalized patients were also associated with other cardiological diseases and comorbidities for which they also received specific medication (medications that also had a secondary hypotensive effect, such as beta-blockers). We must not forget that anticoagulants were not missing from all anti-COVID-19 treatment schemes, because the COVID-19 virus was also responsible for an increased blood coagulability in patients with respiratory infection. Consequently, we can think that the increased blood viscosity could be considered as having an important role, even secondary, in the mechanism and pathogenesis of hypertension.

Regarding dexamethasone administration, as an immune modulator in anti-COVID-19 treatment schemes, we believe that the hypertensive effect did not find expression in the influence of BP values in our patients, since the doses were low–moderate, with only 1 administration per day and for a limited time period of 10 days. BP values of the patients in our study (238 hypertensive patients with SARS-CoV-2) were measured on the even in 14th day of hospitalization which was day of discharge deoarece the European protocol on that date stipulated that patients must be treated in hospital 14 days, considering that from the 14th day they are no longer contagious, regardless of whether they are negative or positive on the RT-PCR test [33]. We consider that the hypertensive effect of dexamethasone can manifest, clinically and cardiovascularly, only in the therapeutic schemes used in hematology, oncology and rheumatology, where the immune-modulating effect of cortisone preparations is obtained at the maximum doses allowed by human pharmacopoeia, as well as in courses of administration that extend over months and even years. However, it is important to note that dexamethasone can have significant cardiovascular effects, including elevating blood pressure and increasing the risk of cardiovascular events. While the risk of these effects may be lower at lower doses and shorter treatment durations, it is still important to monitor blood pressure and other vital signs during treatment with dexamethasone and to adjust treatment as needed.

As for this study’s limitations, we mention that since it was conducted in one hospital only, generalized conclusions are not possible to obtain; moreover, the sample did not allow us to perform a statistical study, the variables being numerous, and the sample itself was not a very large one according to the number of hospitalizations. Additionally, a more thorough comparison between this study’s results and other studies from other hospitals should be done.

## 5. Conclusions

In the SARS-CoV-2 pandemic, HBP was the main reason for hospitalization in the cardiology clinic for COVID-19-positive patients. The classic antihypertensive treatment scheme (diuretics, CCB, ACE and ARB) was sufficient for a slow decrease in BP values from the first 2–3 days after admission and treatment, as well as to maintain normal levels until discharge. There were no paroxysmal BP increases and no urgent hypertensive complications that could be attributed to the dexamethasone treatment. The high percentage of elevated BP values at admission can be attributed to the stress caused by the diagnosis of SARS-CoV-2 infection and the prospect of a subsequent unfavorable evolution. The hypercoagulability associated with COVID-19 viral infection could also be incriminated in the mechanism of the increased BP values upon admission. It is possible that the administration of dexamethasone in high doses and over a longer period is responsible for the known hypertensive effect of this drug, but this situation was not recorded in our patients with SARS-CoV-2. Thus, on the contrary, the measurements of blood pressure values were decreased.

## Figures and Tables

**Figure 1 healthcare-11-01399-f001:**
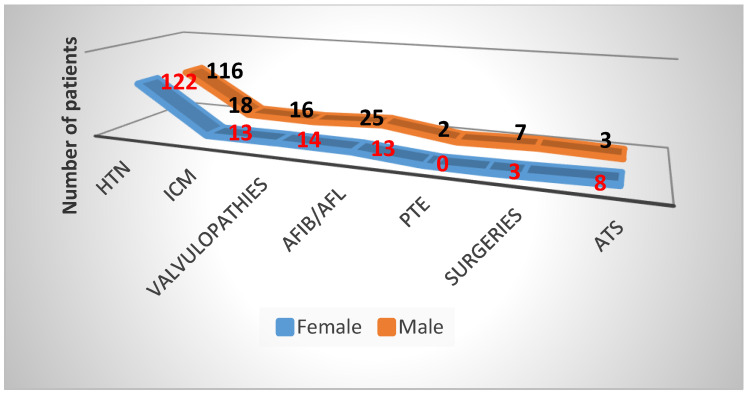
Cardiac diseases. Reason for hospitalization in cardiology unit in COVID-19-positive patients of the study group. HTN, hypertension; ICM, ischemic cardiomyopathy; AFIB/AFL, atrial flutter/atrial fibrillation; PTE, pulmonary thromboembolism; ATS, atherosclerosis).

**Figure 2 healthcare-11-01399-f002:**
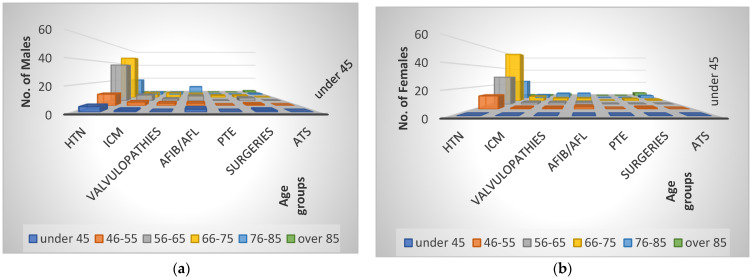
Cardiac diseases. Reason for hospitalization in cardiology unit by age and male (**a**) and female (**b**) gender.

**Figure 3 healthcare-11-01399-f003:**
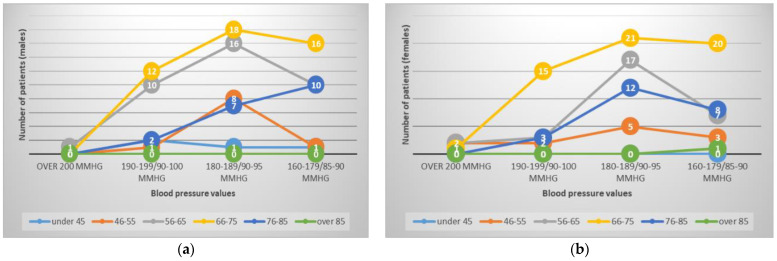
Blood pressure (BP) values at admission, graphically represented by age and sex: (**a**) BP values for men; (**b**) BP values for women.

**Figure 4 healthcare-11-01399-f004:**
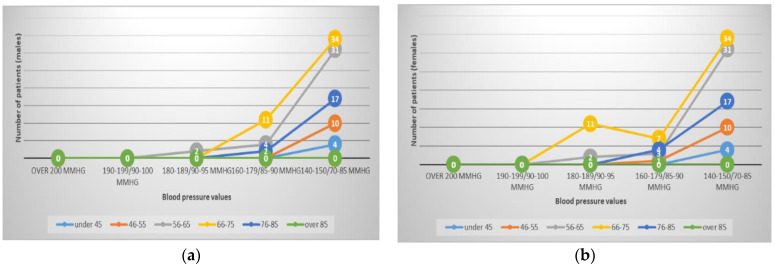
BP values on the 2nd day of admission, graphically represented by age and sex: (**a**) BP values for men; (**b**) BP values for women.

**Figure 5 healthcare-11-01399-f005:**
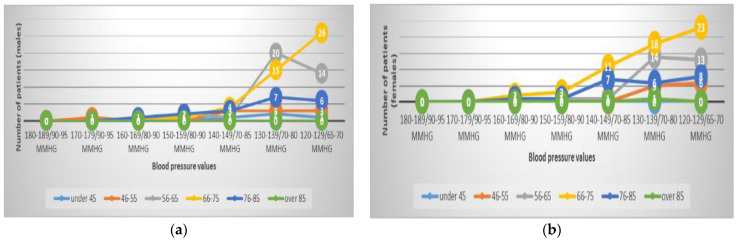
BP values on the 3rd day of admission, graphically represented by age and sex: (**a**) BP values for men; (**b**) BP values for women.

**Figure 6 healthcare-11-01399-f006:**
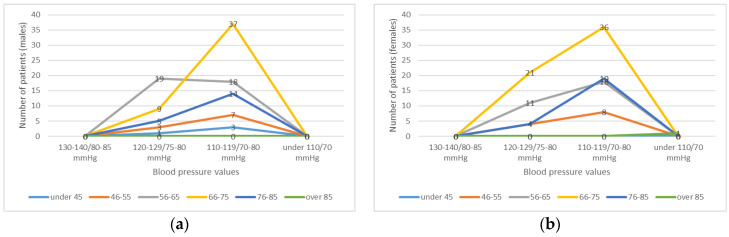
BP values on the 7th day of admission, graphically represented by age and sex: (**a**) BP values for men; (**b**) BP values for women.

**Figure 7 healthcare-11-01399-f007:**
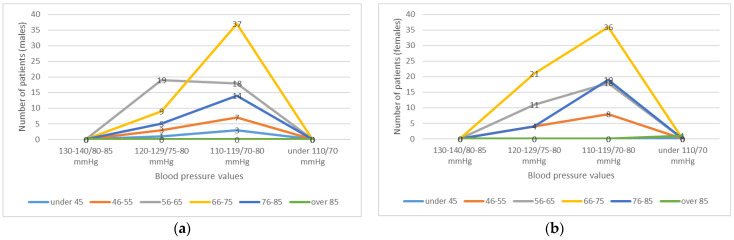
BP values on the 14th day of admission, graphically represented by age and sex: (**a**) BP values for men; (**b**) BP values for women.

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
