# Peer review of "How Dexamethasone Used in Anti-COVID-19 Therapy Influenced Antihypertensive Treatment in Patients with SARS-CoV-2"

_healthcare, 2023, doi:10.3390/healthcare11101399_

Round 1

Reviewer 1 Report

The current manuscript concerns a study of the influence of dexamethasone for Covid 19 therapy in antihypertensive treatment. It is overall interesting, but lacks in some aspects regarding manuscript form and construction, that should be improved. I suggest these modifications before acceptance for publication:

- There are some errors along the manuscript, for example in the keyword list there are parts highlighted in green; english language should also be overall improved;

- “Figure 1’.” should be “Figure 2”, and hence all other figure numbers should also be corrected;

- “Figure 1’.” is also not very perceptible, the type of graph should be changed so that the information is more easily interpreted;

- References are missing for the information included from lines 82 to 94;

- Point 3. in line 94 should include more information as to what this theory is all about, just like done for points 1 and 2;

- The information contained in lines 95 to 96 should be in the methods section;

- Figure captions should include the meaning of the abbreviations that they contain;

- All figures should have improved quality;

- The study´s limitations should be mentioned, since it was done in 1 hospital only, and hence generalized conclusions are not possible to obtain; additionally, a more thorough comparison between this study’s results and other studies from other hospitals should be done.

Author Response

Thank you for taking the time to review the article, for reporting errors in the article and for the suggestions made. Thank you for taking the time to review the article, for reporting errors in the article and for the suggestions made.
I corrected the reported errors, improved the graphics and renumbered the figures.

Reviewer 2 Report

Thank you for the opportunity to review the current manuscript. This manuscript may be of interest to readers. However, the presentation needs to be improved. As in the COVID-19 pandemic, hypertensive crisis complicated treatment in some patients, this manuscript can answer some questions about hypertension severity and treatment response. But before any request, several points should be answered.

·       Please revise the first paragraph of the introduction section,

·       Please correct Dexamethasone to dexamethasone in page 2, line 61.

·       Please bring reference for the following sentence “Paracetamol was also used in anti-covid therapeutic management throughout the pandemic, and was classified in "other therapeutic measures" class, along with antitussives, broncholytics and bronchodilators”

·       Please correct reference through the manuscript. Some cited with space and some without space  for example  along with antivirals, antibiotics, nonsteroidal anti-inflammatory drugs and anti-56 coagulants [1-3].In the present study are listed only drugs that were available and used 64 in the EU and respectively in Romania[10].

Method section

·       The method section can be more clear for better understanding

·       Classification of hypertension, for example 160-179, 180-199,.., should be described in the method section

Results

·       Use table for demographic data

·       The result section is boring, please revised it.

·       Authors used some abbreviations that were not described

·       Data regarding home medications can be summarized in a table

·       It is better to bring some sentence like Hospitalized patients immediately received the medication from the anticovid pro (page 6, line 214) and the next sentence in the method section. In addition, some sentence should be disscused in the discussion section, for example page 8, line 270-273

Author Response

Thank you for taking the time to review the article, for reporting errors in the article and for the suggestions made.
The corrections and suggestions requested by you have been applied to each section of the article.

Reviewer 3 Report

The topic of the manuscript can be of interest, as the authors explain Dexamethasone can pose problems in patients with hypertension.

However, the work has important flaws, as the patients are treated with diverse drugs, and therefore there is not homogeneity. 

The diverse treatment that the patients are receiving for the hypertension or the COVID infection can have different effects in the arterial tension.

The authors have not used any statistics tool to interpret their results and therefore analyze in a consistent way the effect of Dexamethasone.

Other issues:

-       Linear graphs are not appropriate for representing the results of the study. 

-       Figure 1’ (I do not understand why they name it 1’),could be plotted as a simple bar graph.

Author Response

Thank you for taking the time to review the article, for reporting errors in the article and for the suggestions made.
The corrections and suggestions requested by you have been applied to each section of the article with the mention that the homogeneity of the patient's medication cannot be achieved, since patients have pathology-specific medication and individually adjusted. It is also mentioned in the text that the antihypertensive medication was reduced, although dexamethasone was administered, as it is known for its hypertensive effects.

Round 2

Reviewer 2 Report

Dear authors, 

Thanks for your email. Unfortunately, there are some points remained that should be corrected before any decision. for example, no statistical analysis was performed. in addition, some points should be corrected. for example in the conclusion section both abbreviation or SARS COV-2 and SARS-CoV2 were used. which are correct? please revise the text accordingly.

Author Response

The study is of a retrospective observational type and we used only elements of descriptive statistics because, contrary to the expectations in which we would normally have faced an increase in blood pressure values, we observed an improvement in them even with high values of steroid treatment. Also, the batch did not allow us to perform another statistical study, the variables being numerous, and the batch itself was not a very large one, according to the number of hospitalizations.

Reviewer 3 Report

The authors should answer the questions in a document indicating the changes done in the manuscript

Author Response

First of all, we thank you for the time and understanding you have given us. We mention that we tried to take into account all your requests, so that:
1. We have redone all the changes requested by the reviewers, with minor drawbacks mentioned.
2. English language was also overall improved;
3. We tried and improved the quality of the figures;
4. Since the previous subscription, we managed to mention the changes in the text, according to the associated "comment text".